# Glycyl-tRNA Synthetase (GARS) Expression Is Associated with Prostate Cancer Progression and Its Inhibition Decreases Migration, and Invasion In Vitro

**DOI:** 10.3390/ijms24054260

**Published:** 2023-02-21

**Authors:** Ealia Khosh Kish, Yaser Gamallat, Muhammad Choudhry, Sunita Ghosh, Sima Seyedi, Tarek A. Bismar

**Affiliations:** 1Department of Pathology and Laboratory Medicine, Cumming School of Medicine, University of Calgary, Calgary, AB T2N 4N1, Canada; 2Departments of Oncology, Biochemistry and Molecular Biology, Cumming School of Medicine, Calgary, AB T2N 4N1, Canada; 3Department of Medical Oncology, Faculty of Medicine and Dentistry, University of Alberta, Edmonton, AB T6G 2R7, Canada; 4Departments of Mathematical and Statistical Sciences, University of Alberta, Edmonton, AB T6G 2J5, Canada; 5Tom Baker Cancer Center and Arnie Charbonneau Cancer Institute, Calgary, AB T2N 4N1, Canada; 6Prostate Cancer Center, Rockyview General Hospital, Calgary, AB T2V 1P9, Canada

**Keywords:** prostate cancer, GARS, PTEN, proliferation, cell cycle regulation

## Abstract

Glycyl-tRNA synthetase (GARS) is a potential oncogene associated with poor overall survival in various cancers. However, its role in prostate cancer (PCa) has not been investigated. Protein expression of GARS was investigated in benign, incidental, advanced, and castrate-resistant PCa (CRPC) patient samples. We also investigated the role of GARS in vitro and validated GARS clinical outcomes and its underlying mechanism, utilizing The Cancer Genome Atlas Prostate Adenocarcinoma (TCGA PRAD) database. Our data revealed a significant association between GARS protein expression and Gleason groups. Knockdown of *GARS* in PC3 cell lines attenuated cell migration and invasion and resulted in early apoptosis signs and cellular arrest in S phase. Bioinformatically, higher *GARS* expression was observed in TCGA PRAD cohort, and there was significant association with higher Gleason groups, pathological stage, and lymph nodes metastasis. High *GARS* expression was also significantly correlated with high-risk genomic aberrations such as *PTEN*, *TP53*, *FXA1*, *IDH1*, *SPOP* mutations, and *ERG*, *ETV1*, and *ETV4* gene fusions. Gene Set Enrichment Analysis (GSEA) of *GARS* through the TCGA PRAD database provided evidence for upregulation of biological processes such as cellular proliferation. Our findings support the oncogenic role of GARS involved in cellular proliferation and poor clinical outcome and provide further evidence for its use as a potential biomarker in PCa.

## 1. Introduction

Prostate cancer (PCa) is the second leading cause of cancer-related deaths and the most common type of cancer in men [1]. With the advances in sequencing and diagnostic technologies, PCa trails other cancers in the field of gene-targeted therapy. Since the prostate-specific antigen test lacks specificity, biomarkers have become an essential tool in diagnosing and assessing prognosis in PCa [2]. However, the use of potential biomarkers such as *ERG*, *PTEN*, and *TP53* help in predicting the outcomes of lethal PCa [2,3,4]. However, there is an urgent need to identify biomarkers that could potentially assess early diagnosis or therapeutic benefits.

Glycyl-tRNA synthetase (*GARS*) is an aminoacyl-tRNA synthetase which is involved with charging amino acids onto their respective tRNA molecules in the primary steps of protein synthesis [5,6]. Wang et al. demonstrated that the increased expression of GARS in hepatocellular carcinoma (HCC) tissue was significantly associated with both poor overall survival and disease-free survival [5]. In vitro, *GARS* knockdown in HCC cells promoted apoptosis, inhibiting HCC cells proliferation and cell cycle [5]. *GARS* was also documented to have increased expression in lung adenocarcinoma and was associated with an unfavorable prognosis [7]. In urothelial carcinoma, Chen et al. displayed the role of GARS as a urine biomarker aiding the diagnosis of urothelial cancer [8]. Furthermore, *GARS* appears to play an oncogenic role in breast cancer. Li et al. exhibited the increased expression of GARS in breast cancer tissue compared to healthy tissue and demonstrated that reduced cellular proliferation, colony formation, and migration abilities were in line with *GARS* inhibition [9]. In addition, GARS appeared to be significantly overexpressed in early-stage breast cancer compared to benign breast disease and normal healthy control samples [10]. Furthermore, a previous study on the role of Aminoacyl TRNA Synthetases showcased the direct interaction between the Androgen Receptor and *GARS* promoter in PCa [11]. The current data point towards the role of *GARS* as a possible oncogene in various cancers. In this study, we explored the prognostic role of *GARS* as a potential biomarker in PCa by assessing its protein expression using IHC and validating its mRNA expression in public cohorts. Additionally, we characterized the role of GARS in vitro cellular models and correlated this to changes in cellular invasion, proliferation, and cell cycle.

## 2. Results

### 2.1. Moderate and High GARS Expression in PCa Patients

GARS IHC showed moderate and high intensity of GARS protein expression in 82/134 (61.2%) and 31/134 (23.1%) of cases, respectively. The mean expression was 1.73 ± 0.59 in benign, 1.98 ± 0.55 in incidental, 2.07 ± 0.68 in advanced, and 2.09 ± 0.68 in castrate-resistant cases (*p =* 0.626) (Figure 1).

### 2.2. GARS Expression in Relation to Gleason Grade Grouping (GG)

High GARS protein expression was seen in 28.6% of GG 5 cases vs. 8.5% of GG 1. Comparatively, low GARS expression was seen in 91.5% of GG 1 cases vs. 71.4% of GG 5 cases (*p* = 0.023). Overall, our data revealed that GARS expression increased dramatically with higher Gleason Groups in this cohort (Table 1).

### 2.3. GARS in Relation to Overall Survival (OS) and Cause-Specific Survival (CSS)

We explored the relationship between *GARS* mutations in relation to overall (OS) and cancer-specific survival (CSS), using 5015 samples in 25 studies in a public database for prostatic adenocarcinomas. Figure 2A,B confirm the prognostic significance of *GARS* genomic alteration and its relation to poor OS and CSS for patients diagnosed with PCa.

### 2.4. GARS mRNA Is Upregualted in Most Cancer Types and Significantly Associated with Poor Prognostic Biomarkers

The Pan-Cancer data analysis revealed that *GARS* mRNA expression was significantly upregulated in all 22 cancer types (Figure 3A). Specifically, in PCa, there was a significant increase in *GARS* expression in tumor tissue compared to normal tissue. This difference was more significant when the tumor was compared to non-cancerous normal samples rather than to adjacent normal tissue of the prostate. Furthermore, the *GARS* gene expression was upregulated significantly in metastatic PCa (Figure 3B–D). Additionally, we found that *GARS* overexpression significantly correlated with lethal disease genomic aberration, such as *PTEN*-loss, *TP53* mutant tumors, *ERG*, *ETV1*, *ETV4* gene fusion, *FXA1*-mutation, *IDH1*-mutation, and *SPOP*-mutation (Figure 3E–G).

### 2.5. GARS Expression Is Associated to Lymph Node and Pathological Staging in PCa Based on TCGA PRAD Database

Bioinformatic analysis using *TCGA PRAD* database revealed that elevated *GARS* expression in PCa is significantly associated with lymph node involvement and higher pathological staging (Figure 4A,B). Furthermore, *GARS* gene expression is significantly associated with the number of lymph nodes involved in metastatic PCa (Figure 4C). Higher *GARS* expression appears to be significantly related to increased residual tumor levels (Figure 4D) and higher Gleason scores in PCa (Figure 4E).

### 2.6. GARS Overexpression Depicted Potential Biological and Molecualar Functions Associated with Poor Clinical Outcomes in TCGA PRAD

Differentially expressed genes were analyzed for *GARS.* Our data uncovered an interesting and distinctive profile of GARS associated genes (Figure 5A). Heatmaps depict the top 50 positively (Figure 5B) and negatively correlated genes (Figure 5C). *GARS* GSEA conveys an interesting pattern of upregulated genes involved in biological processes, cellular components, and molecular functions (Figure 5D). Positively correlated genes for molecular function were mostly involved in protein binding (6492 genes), nucleic acid binding (2363 genes), and nucleotide binding (1308 genes). Cellular components in GARS upregulated group revealed enrichment of genes localized to membrane (4767 genes) and the nucleus (4154 genes), which appears to contain the most altered gene in cellular components. Lastly, our data demonstrated the upregulation in biological processes, such as genes involved in biological regulation (6513 genes), metabolic processes (6252 genes), cellular component organization (3649 genes), cellular proliferation (1184 genes), and reproduction (784 genes) (Figure 5D).

### 2.7. GARS Overrepresentation Enrichment Analysis (ORA) in PCa TCGA PRAD Database

Two separate enrichment methods, GSEA and ORA, were performed for this analysis. When overrepresentation genes in association to *GARS* were analyzed using ORA, many biological processes such as translation initiation, mitochondrial gene expression, and DNA strand elongation were identified (Appendix A). Furthermore, genes involved in cellular components, such as in condensed chromosome, mitochondrial protein complex, and replication fork, were seen (Appendix A). Lastly, genes involved in molecular functions such as unfolded protein binding, cyclin-dependent protein kinase activity, ligase activity, and catalytic activity acting on DNA were further overrepresented (Appendix A). Our GSEA *GARS* analysis provides evidence for the overrepresentation of genes in biological processes such as RNA processing, cellular response to stress, negative regulation of gene expression, cell development, and cell cycle (Appendix A). Genes involved in molecular functions such as structural constituent of ribosome, RNA binding, and kinase binding were also seen (Appendix A).

### 2.8. GARS Expression in PCa Cell Lines and GARS Knockdown

GARS protein expression levels were estimated in PC3, PC3-ERG, and DU-145 cell lines using Western Blot. Our data showed high GARS protein expressions in PC3, PC3-ERG, and DU-145 (Figure 6A).

*GARS* knockdown was successfully performed on PC3 and PC3-ERG cell lines. The optimal duration of the knockdown was observed to be nearer 36 h than to to 48 h. Knockdown efficiency was further tested using Western Blot analysis (Figure 6B). Furthermore, we observed a significant downregulation of Cyclin B1, P-PDK1, and AURKA and AKT after *GARS* knockdown in PC3 cell lines, but there was no significant difference observed in PC3-ERG cells. (Figure 6C). These data suggested a potential role of GARS in cell cycle and proliferation.

### 2.9. GARS Knockdown Attenuated PCa Cell Migration and Invasion

To further elucidate the role of *GARS* as an oncogene in PCa, we examined the cells migration and invasion using transwell assays. *GARS* knockdown significantly reduced the ability of PC3 and PC3-ERG cells to invade and migrate in vitro (Figure 7).

### 2.10. GARS Is a Cell Cycle Regulator in PCa

We explored the potential role of *GARS* as a cell cycle regulator and its involvement in the proliferation of PCa cell lines. Our data revealed significant dysregulation of cell cycle after knockdown of *GARS* using flowcytometry. We found that after the knockdown, the cells are arrested in the S phase and are unable to enter the mitotic phase. Furthermore, the cells appear to undergo early apoptosis after *GARS* knockdown when compared to the negative control (Figure 8).

## 3. Discussion

PCa is a heterogenous disease with a high overall survival for localized disease. However, the percent survival at 5 years decreases to 26–30% for advanced and metastatic castrate-resistant PCa [12,13]. To overcome this issue, the search for other biomarkers with easily accessible and reliable outcomes is urgently needed.

In the current study, we reported that *GARS* overexpressed significantly in PCa and 21 other cancers. In our cohort, we also found that GARS was significantly upregulated in localized, incidental, advanced, and metastatic PCa tumors when compared to normal tissue. From the TCGA PRAD database, GARS overexpression was correlated with the presence of commonly occurring oncogenic mutations. These included *PTEN* loss *P53* mutant tumors, *ERG*, *ETV1*, *ETV4* gene fusion, *FXA1*-mutation, *IDH1*-mutation, and *SPOP*-mutation. Most of these mutations are currently used as a clinical biomarker to determine the disease prognosis and outcomes. Interestingly, their association with GARS could suggest that it is an integral part of producing a more lethal phenotype in tandem with these mutations [14,15]. Previous studies in HCC indicated that GARS overexpression was significantly associated with poor overall and disease-free survival [5,9]. Similarly, looking at TCGA PRAD database, we found that *GARS* overexpression is significantly related to higher PCa pathological stages, Gleason grade groups, and lymph nodes metastasis. Furthermore, our clinical data showed a significant association between high *GARS* expression and higher Gleason grade grouping. Our IHC results demonstrated trends of increased protein GARS expression from benign, to incidental, advanced, and castrate-resistant PCa samples. It is important to note that clinicians use Gleason grade grouping, lymph node involvement, and residual tumor as means of assessing PCa patients’ progression risk and prognosis [16,17,18,19]. Furthermore, it appears that *PTEN* loss has been associated with poor outcome in localized and castrate-resistant PCa [20,21]. *TP53* has been shown to predict Abiraterone/Enzalutamide outcomes in metastatic castrate-resistant PCa [22]. Herein, we found that *GARS* gene overexpression is significantly associated with *PTEN* loss, ERG gene fusion and *TP53* mutational status among additional genomic aberrations. This suggests that GARS exhibit oncogenic effects and may be of potential use as a prognostic biomarker in lethal PCa. Together, our results provide clinical and molecular support for the role of *GARS* as an oncogene and possible biomarker in PCa.

Through our investigation of the mechanism underlying the oncogenic role of *GARS* we performed gene set enrichment analysis of GARS overexpressed/overrepresented cases using GSEA and ORA analysis on the *TCGA PRAD* database, we concluded that many tumor-associated process are upregulated when *GARS* is overexpressed. For example, we found that genes involved in biological processes such as cellular proliferation, reproduction, biological regulation, translation initiation, DNA strand elongation, and metabolic processes were enriched in GARS-overexpressed cases. Additionally, genes involved in molecular functions such as cyclin-dependent kinase activity and catalytic activity acting on DNA were also upregulated. Previous studies performed on the gene ontology analysis of *GARS* indicated that many genes involved in cell division, cell proliferation, and cell cycle were enriched [5]. Our data indicate that *GARS* might be involved in PCa cell cycle regulation and proliferation. Furthermore, using cellular in vitro models, we documented that *GARS* knockdown inhibits the migration and invasion abilities of PCa cells. Previously, *GARS* overexpression has been shown to accelerate cell cycle, migration, and invasion of breast cancer cells [5,9]. Mechanistic studies also indicated that *GARS* may act as an oncogene in breast cancer through controlling the mTOR pathway and regulating cellular proliferation [9]. Our data also suggest that *GARS* knockdown results in S phase arrest and promotes early apoptosis which attenuates PCa cellular proliferation. This is further supported by significant downregulation of Cyclin B1 and AURKA cell cycle regulators after *GARS* knockdown [23]. Our data also suggest a downregulation in the levels of Pyruvate Dehydrogenase Kinase 1 (PDK1) in association with *GARS* knockdown. PDK1 overexpression has been shown to induce proliferation and metastasis through the Warburg effect in non-small cell lung cancer [24,25]. Furthermore, PDK1 knockdown in vitro has been shown to reduce PCa cellular proliferation, migration and invasion in vitro in PCa [26]. These data point towards the oncogenic ability of *GARS* in connection with PDK1 in regulating PCa proliferation in vitro. The limitations of this study include its lack of in vivo support for the down regulations of *GARS* in animal models, which would further elucidate the oncogenic ability of *GARS* in tumors. Further research is required to elucidate the mechanism of *GARS* in proliferation of PCa.

## 4. Materials and Methods

### 4.1. Study Population, Tissue Microarray Construction, and Pathological Analysis

A tissue microarray (TMA) was constructed from a cohort of 264 patients diagnosed with adenocarcinoma of the prostate. GARS expression in association to Gleason grade groups was assessed. Histological diagnoses of individual cores on the TMA were confirmed by the study pathologist (TAB). GARS intensity expression was assessed using a four-tiered system (0, negative; 1, weak; 2, moderate; 3, high intensity). Gleason grade grouping were assessed according to the 2018 WHO and ISUP grade group by the study pathologist (TAB).

### 4.2. Immunohistochemistry (IHC)

GARS protein expression was assessed using IHC on the Dako Omnis auto Stainer. Briefly, 4 µm formalin-fixed paraffin-embedded (FFPE) sections were first treated with citrate epitope retrieval buffer (pH 6.0). Following that, incubation with rabbit monoclonal GARS antibody (1:500) (Cat#HPA019097, Sigma-Aldrich, St. Louis, MO, USA) was used. After the secondary antibody incubation, the FLEX DAB+ Substrate Chromogen system was used as a detection reagent.

### 4.3. Cell Lines

Human PCa cell lines used in this study include LnCaP, PC3, and DU-145. All cell lines were purchased from the American Type Culture Collection (ATCC; Manassas, CA, USA). Stable PC3-ERG was obtained from Felix Feng, University of Michigan [27].

DU-145 cells were cultured in DMEM media (GIBCO life technology, Grand Island, NY, USA). PC3 and PC3-ERG cells were cultured in DMEM/F12 (GIBCO life technology, Grand Island, NY, USA). LnCaP cells were cultured in RPMI 1640 medium (GIBCO life technology, Grand Island, NY, USA). All the above were supplemented with 10% FBS (GIBCO life technology, Grand Island, NY, USA) and grown at 37 °C in 5% CO_2_ environment.

### 4.4. Cell Line Transfection via RNA Silencing

GARS knockdown was performed using pre-designed silencer siRNA *GARS*, and scramble siRNA was used as a negative control (Cat# AM16708, Ambion, Grand Island, NY, USA). PC3 and PC3-ERG cells were plated in six well plates until 75–80% confluency was reached. Furthermore, the siRNA transfection mix, including Opti-MEM (Cat#31985-070, GIBCO life technology, Grand Island, NY, USA) and Lipofectamine RNAiMAX (Cat# 13778-075, Invitrogen, Carlsbad, CA, USA), were used according to the manufacturer’s instructions. The transfection efficiency was checked by western blotting.

### 4.5. Western Blot

Total protein was extracted using RIPA buffer (Sigma-Aldrich, St. Louis, MO, USA) pre-mixed with protease inhibitors and PMSF (Cat# 5872S, Cell signaling, Danvers, MA, USA). When loading on the polyacrylamide SDS gel, equal quantities of proteins were loaded in each separate lane. PVDF membrane (Cat# ISEQ85R, Millipore Sigma, Burlington, MA, USA) was used for the transfer of the proteins. The membrane was placed in blocking buffer prepared with 5% skim milk in PBS for 1 h at room temperature. After blocking, the membrane was incubated with primary antibodies (Appendix A) overnight at 4 °C with shaking. After primary antibody incubation and washing, the membrane was incubated with either anti-mouse IgG or anti-rabbit IgG secondary antibody conjugated to HRP horseradish peroxidase (Cell signaling, Danvers, MA, USA) in TBS for 1 h. at 37 °C. After final washings, the signal was detected using ChemiDoc imaging system (Bio-Rad Laboratories, Hercules, CA, USA).

### 4.6. Migration and Invasion Assay

PC3 and PC3-ERG cells were seeded in six well plates. They were transfected with GARS siRNA#1, siRNA#2, or scramble siRNA (negative control). After 24 h. post transfection, cells were trypsinated and counted with automatic cell counter (Olympus, PA, USA). Approximately 25,000 cells were placed on either 0.8 µ insert Corning Biocoat control inserts for migration assay (Ref# 354578, Corning, Bedford, MA, USA) or Corning Matrigel invasion chamber (Ref# 354480, Corning, Bedford, MA, USA). After 48 h., all cells were fixed and stained with Diff Quick (Siemens Healthcare diagnostics, Tarrytown, NY, USA). All cells were captured on brightfield 10× and 40× magnification using an inverted EVOS FL life microscope. The number of cells for multiple frames were counted for each treatment and average from the 40× magnification. The knockdown was compared to the negative control.

### 4.7. Flow Cytometry

For cell cycle, GARS knockdown using siRNA#1, siRNA#2, and a scrambled siRNA used as negative control with appropriate replicates were prepared as previously stated. The cells were harvested after the knockdown, washed with cold PBS, and fixed in 70% ethanol for at least 2 h. Further, they were stained with 100 µg/mL of RNase A in PBS and 50 µg/mL propidium iodide (Cat#F10797, Invitrogen, Carlsbad, CA, USA). The DNA content of cells were analyzed using BD LSR II Flow Cytometer. For apoptosis or Annexin V/PI assay, cells were grown and transfected as described in MM 2.3 above. The cells were further trypsinated and treated with Annexin V apoptosis kit (Cat# V13241, Invitrogen, Carlsbad, CA, USA) per the manufacturer’s instructions. All data were analyzed using FlowJo™ v10 Software-BD Biosciences.

### 4.8. GARS Expression in the Cancer Genome Atlas Prostate Adenocarcinoma (TCGA PRAD)

Genomic signature data were obtained from TCGA PRAD transcriptomics database [28]. Pan-cancer analysis was used across TCGA, GTex, and TARGET databases to analyze *GARS* expression in 22 types of tissues, and in tumor vs normal. This tool functioned based on the RNA-seq-rapid analysis servers. *GARS* gene expression for tumor vs. adjacent normal tissue and tumor vs non-adjacent normal tissue was further analyzed (R0 = no residual tumor, R1 = microscopic residual tumor, R2 = macroscopic residual tumor [29]. Results were blotted and gene expression at tumor was compared to normal at each of the quantile cut-off values (minimum, 1st quartile, median, 3rd quartile, maximum) [28]. Furthermore, we used the data available in TCGA PRAD database to compare mutations of *PTEN*, *TP53*, *FXA1*, *IDH1*, *SPOP*, and gene fusions such as *ERG*, *ETV1*, and *ETV4* with *GARS* expression. Furthermore, we compared *GARS* expression to PCa pathological stage, lymph node involvement, number of lymph nodes involved, and residual tumor in this database.

We used LinkedOmics (http://www.linkedomics.org (accessed on 12 December 2022)) to explore GSEA and ORA functions of *GARS* utilizing the TCGA PRAD database [30]. *GARS* overrepresented and gene set enrichment were analysis and ranked based on highest FDR score. Furthermore, they were grouped into molecular functions, biological processes, and cellular components using WEB-based Gene Set Analysis Toolkit and Explorer [31].

### 4.9. GARS Expression in TCGA PRAD Analyzed through UALCAN

Data collected from TCGA were analyzed through UALCAN (http://ualcan.path.uab.edu/index.html (accessed on 12 December 2022)) in order to explore the relationship between *GARS* transcript to pathological features, such as Gleason score and association to various gene mutations [32,33]. *GARS* expression was compared to Gleason scores 6–10 and normal. Furthermore, *GARS* expression was compared in normal samples, *TP53* mutant tumors, and *TP53* wild type tumors. Box–whisker plots contain the minimum, 1st quartile, median, 3rd quartile, maximum, and interquartile range. Welch’s *T*-test was used to analyze the difference in expression levels between normal and tumors.

### 4.10. Statistical Analysis

Descriptive statistics were used to describe the current study data. For categorical data, frequency and proportions were reported. For continuous data, mean and standard deviations were reported. A two-tailed *t*-test was used to compare two continuous measures (*p*-value < 0.05). Box–whisker plots contain the minimum, 1st quartile, median, 3rd quartile, maximum, and interquartile range. Overall survival (OS) was defined as the time from diagnosis to death. Analysis was performed using Graph pad version 7. *p*-value < 0.05 was used for statistical significance and two-sided t tests were conducted.

## 5. Conclusions

In conclusion, our study provides evidence for the oncogenic role of *GARS* in PCa. We documented that *GARS* is overexpressed in various cancers, including PCa, and is associated with a higher pathological stage and number of lymph nodes involved. Furthermore, GARS overexpression is further associated with higher Gleason grade groups, as well as patients’ clinical prognosis. Furthermore, *GARS* knockdown reduces the ability of PCa cells to invade and migrate while inducing S phase arrest. Our data suggest that *GARS* functions as a cell cycle and proliferative regulator in association with PDK1 in PCa. Further research is needed to demonstrate the mechanisms underlying *GARS* in the proliferation pathways in PCa and other types of cancer.

## Figures and Tables

**Figure 1 ijms-24-04260-f001:**
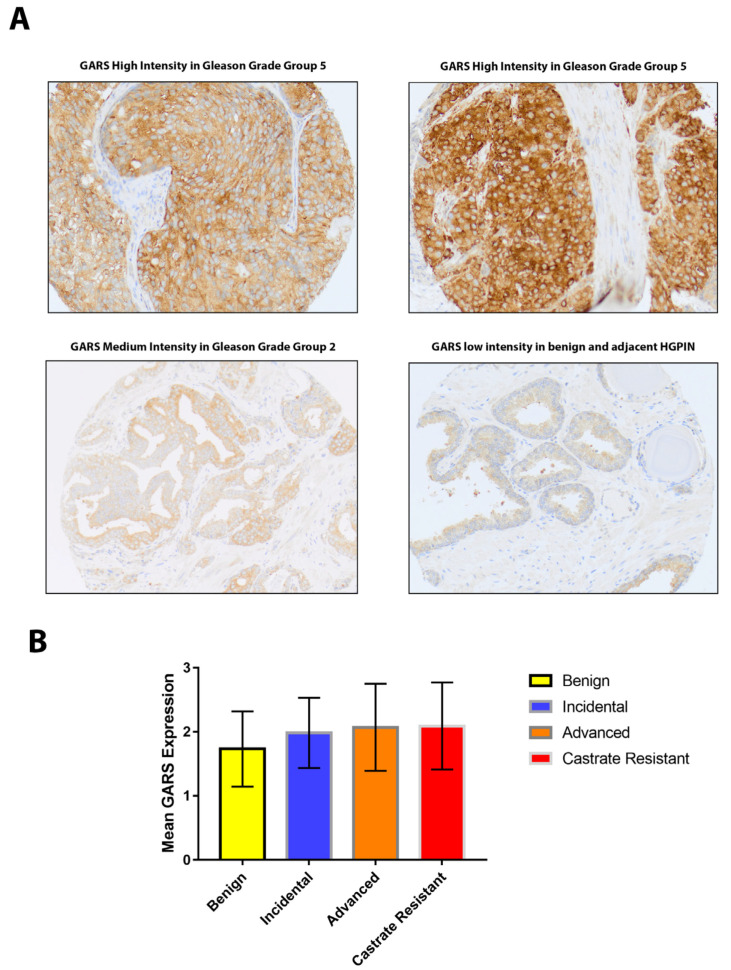
GARS protein expression in prostate tissue. (**A**) IHC staining of GARS showing low intensity in benign and adjacent HGPIN (high grade prostatic intraepithelial neoplasia) (bottom right panel, 20× magnification), medium intensity in prostate cancer Gleason grade Group 2 (top left panel, 20× magnification), and high intensity in prostate cancer Gleason grade Group 5 (left and right bottom panel) (20× magnification). (**B**) Box plot demonstrating mean expression of GARS in benign (*n* = 44), incidental (*n* = 60), advanced (*n* = 69), and castrate-resistant PCa (*n* = 63). Errors bars indicated ± standard deviation (SD). One-way ANOVA was performed, *p* = 0.626.

**Figure 2 ijms-24-04260-f002:**
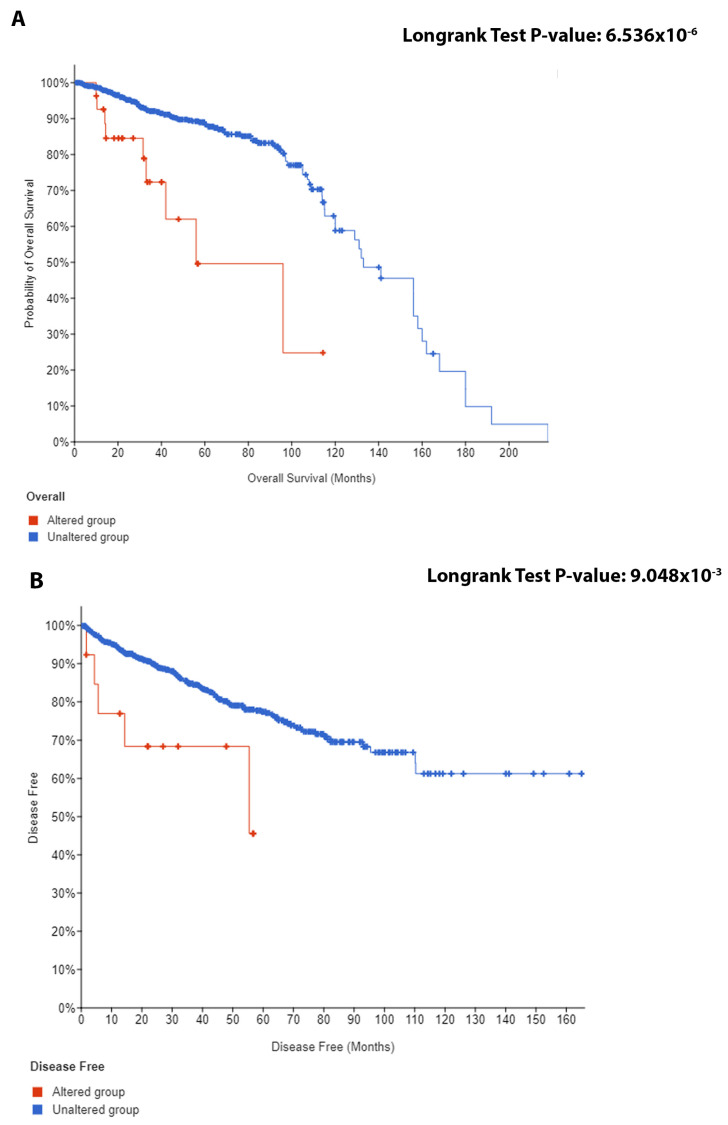
The Kaplan–Meier (KM) curves for *GARS* genomic alteration for (**A**) PCa overall survival and (**B**) PCa disease-free survival. Data were obtained from cBioportal (contains 5015 samples). Genomic alteration includes any of the mutations (Missense, in frame, truncating, other), structural variant/fusion, and copy number alteration in the *GARS* gene.

**Figure 3 ijms-24-04260-f003:**
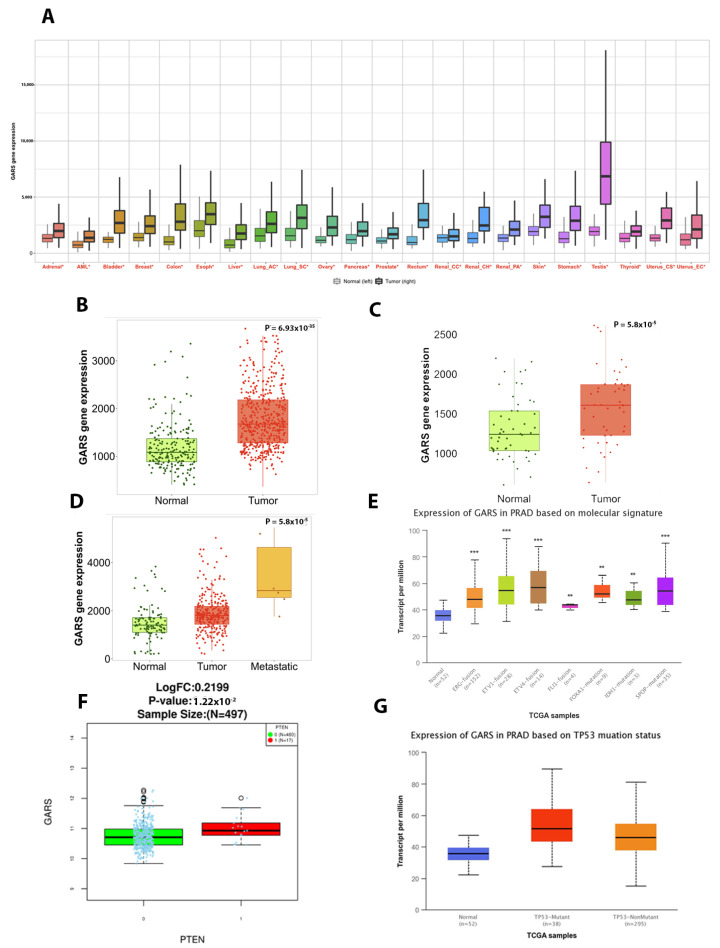
Gene expression analysis of *GARS* in tumors and normal tissues. (**A**) Boxplots represent the *GARS mRNA* gene expression in 22 types of cancer in Tumor (right) and normal (left). The Mann–Whitney U test was used, and significant difference is marked with red * (*p* < 0.01). (**B**–**D**) Boxplots represent the *GARS* RNA-seq gene expression in (**B**) PCa tumor and normal tissue from non-cancerous patients, (**C**) Pca tumor and paired adjacent-normal tissue, (**D**) Pca tumor, normal tissue, and metastatic Pca. (**E**) Boxplots showing *GARS* association with common PCa mutations (*ERG*, *ETV1*, *ETV4*, *FLI1*, *FOXA1*, *IDH1* and *SPOP*) (Asterisks indicate significant *p* value, * *p* value < 0.05, ** *p* value < 0.01, *** *p* value < 0.0001). (**F**) Boxplots representing *GARS* gene expression in relation to *PTEN* loss (0—wild type, 1—loss mutations). (**G**) Boxplot shows *GARS* gene expression in relation to *TP53* mutations (red represents mutated gene and green is wildtype).

**Figure 4 ijms-24-04260-f004:**
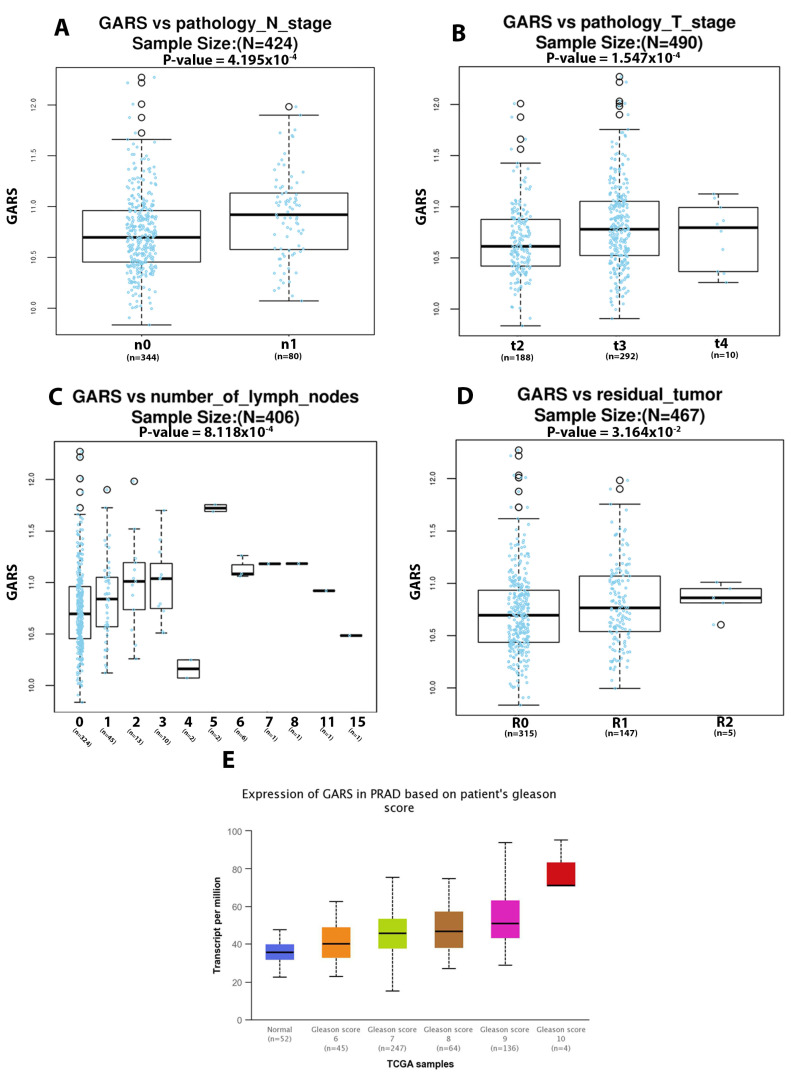
Gene expression analysis of *GARS* in relation to clinical features from TCGA PRAD database. Boxplots represent *GARS RNA-seq* gene expression in relation to (**A**) pathological nodal stage involvement assessed using Wilcox Test (n0 = no cancer in lymph nodes, n1 = cancer in lymph nodes), (**B**) pathological tumor stage assessed using Kruskal–Wallis Test (Stage 2, Stage 3, Stage 4), (**C**) number of lymph nodes involved assessed using Kruskal–Wallis Test, and (**D**) residual tumor levels assessed using Kruskal–Wallis test (R0 no residual tumor, R1 = microscopic residual tumor, R2 = macroscopic residual tumor. (**E**) Gleason score assessed via Welch’s *T*-Test. *p*-value for normal versus Gleason score 6, 7, 8, 9 (2.57 × 10^−2^, 2.12 × 10^−12^, 9.82 × 10^−8^, 1.11 × 10^−16^, respectively).

**Figure 5 ijms-24-04260-f005:**
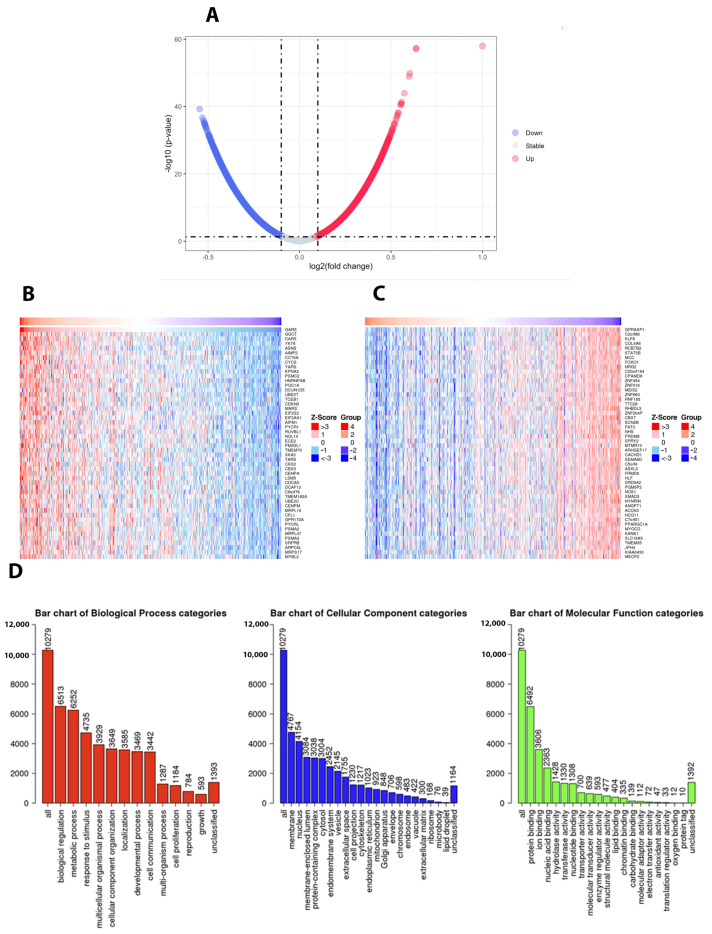
*GARS* Gene Set Enrichment Analysis in PCa *TCGA PRAD*. (**A**) Volcano blot showing the differentially expressed genes associated with *GARS.* Downregulated genes (blue), unchanged genes (grey), upregulated genes (red). The horizontal dashed line represents the Log2 threshold. (**B**) Heatmaps indicating the top 50 positively correlated genes. (**C**) The top 50 negatively correlated genes in relation to *GARS* overexpression. (**D**) Bar blots indicating the GSEA analysis categories, including biological functions (Red), cellular components (Blue), and molecular functions (Green). *FDR* was calculated using the Benjamini–Hochberg test, and height of the bar indicated the number of IDs in the user list.

**Figure 6 ijms-24-04260-f006:**
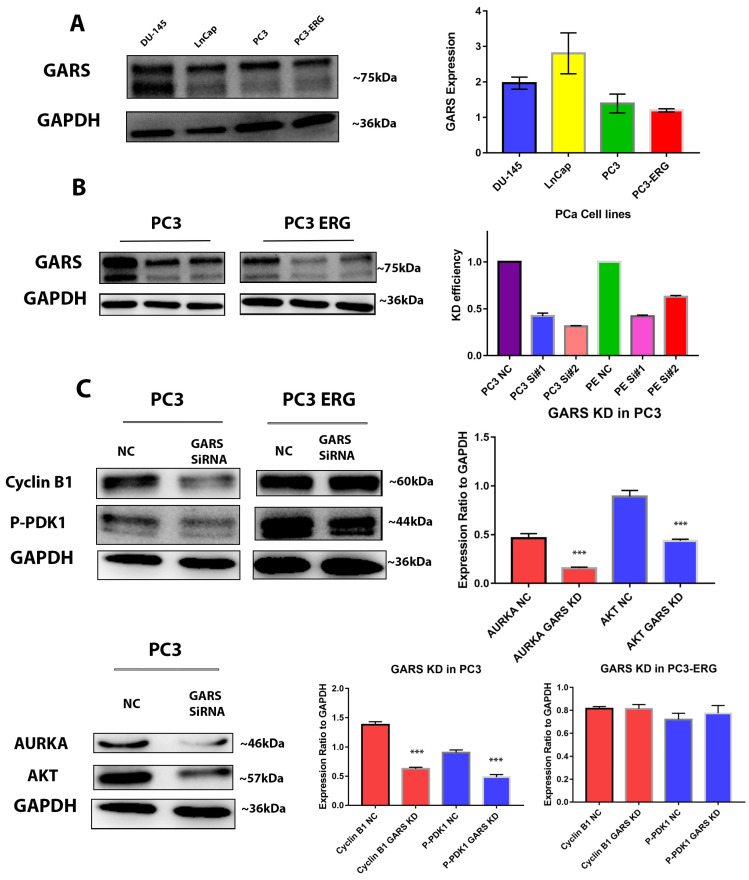
Western blot analysis of GARs in PCa cell line. (**A**) Western blot of GARS expression in PCa cell lines: DU-145, LnCap, PC3, and PC3-ERG cells. (**B**) *GARS* Knockdown efficiency in PC3 and PC3-ERG cell lines using *GARS* siRNA#1, #2, and scrambled siRNA as negative control after 48 h. (**C**) Western blotting analysis of Cyclin B1, P-PDK1, AURKA, and AKT proteins expression in *GARS* knockdown PC3 and PC3-ERG cells compared to negative control cells. GAPDH was used as internal housekeeping control (*** Asterisks indicate significant *p* value < 0.0001).

**Figure 7 ijms-24-04260-f007:**
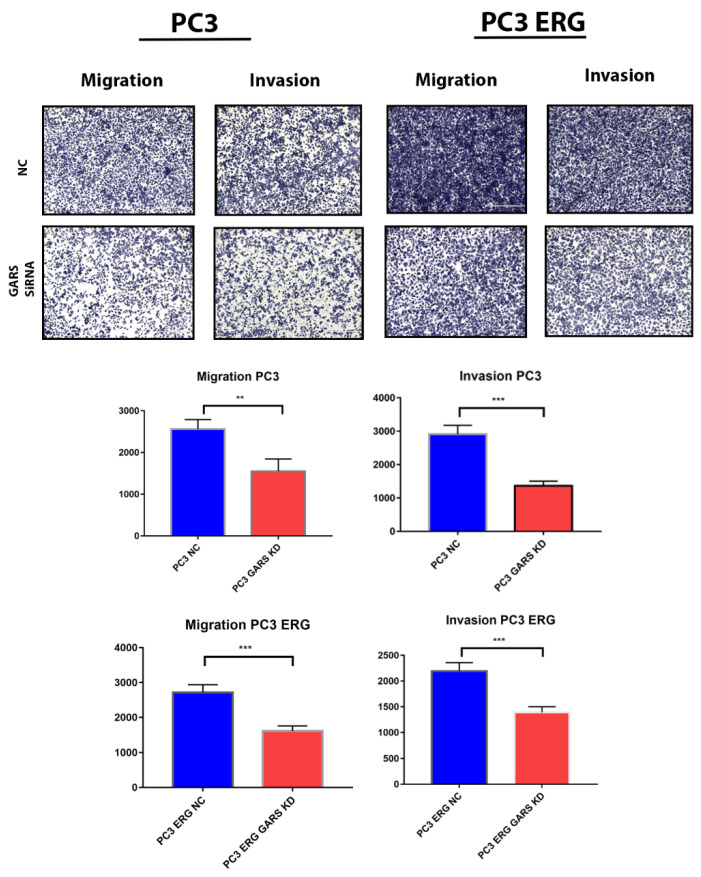
*GARS* knockdown reduced the ability of PC3 and PC3-ERG cells to migrate and invade in vitro analyzed over 3 replicates (Asterisk *** indicates *p* < 0.001 and ** indicates *p* < 0.01). Scale bar in the figure represents 400 μm.

**Figure 8 ijms-24-04260-f008:**
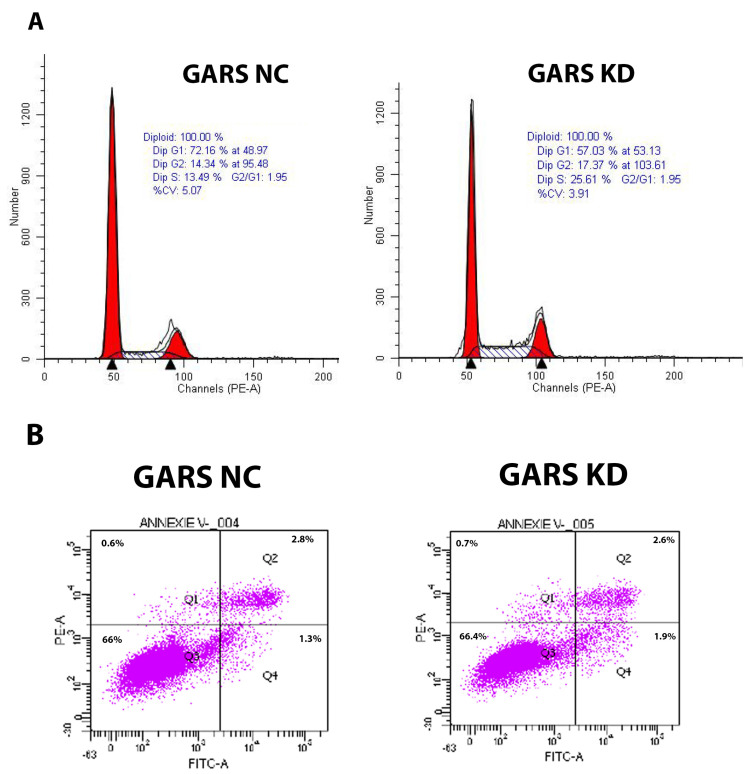
Cell cycle and apoptosis assay analysis post *GARS* knockdown in PC3 cell line. (**A**) Cell cycle analysis using PI staining (chi square test; *p* < 0.001). (**B**) Apoptosis analysis using Annexin V. Data analyzed using FACSDiva. Cell cycle using PI staining and apoptosis assays after *GARS* knockdown (chi square test; *p* < 0.001). Data analyzed using FACSDiva.

**Table 1 ijms-24-04260-t001:** *GARS* expression in relation to patient’s Gleason Grade Groups.

GARS	Gleason Grade Group 1	Gleason Grade Group 2	Gleason Grade Group 3	Gleason Grade Group 4	Gleason Grade Group 5
Score 0, 1, 2	54 (91.5%)	9 (60.0%)	15 (78.9%)	9 (81.8%)	55 (71.4%)
Score 3	5 (8.5%)	6 (40.0%)	4 (21.1%)	2 (18.2%)	22 (28.6%)

Negative—0, weak—1, moderate—2, high—3.

## Data Availability

RNAseq data was extracted from the open source The Cancer Genome Atlas (*TCGA)* Prostate Adenocarcinoma (*PRAD*) database. Genome alteration data was extracted from cBioportal.

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
