# Peer review of "Glycyl-tRNA Synthetase (GARS) Expression Is Associated with Prostate Cancer Progression and Its Inhibition Decreases Migration, and Invasion In Vitro"

_ijms, 2023, doi:10.3390/ijms24054260_

Round 1
Reviewer 1 Report
Comment: This is an interesting study and the authors have collected a unique dataset using cutting edge methodology. The paper is generally clear, and structured. Sufficient information about the previous study findings is presented for readers to follow the present study rationale. The methods are generally appropriate, although clarification of a few details and provision of a rationale for the use of this particular method of measurement. In several instances, this article has some shortcomings in regard to this text, such as the table should be addressed in more detail from your results or make another table for clear your discovery in avoiding vague and one-to-one correspondence with text.
Specific comments follow,
Abstract
Gleason Groups (Line23, 26) “g”roups--- lowercase or abbreviation GG for Gleason Groups.
Materials and Methods
2.8. GARS expression in TCGA PRAD
Please show full name (TCGA PRAD) for the first time and link in Line 134 (not only the reference 12 & “Data Availability Statement” Line 378-379).
2.9. GARS expression in TCGA PRAD analyzed through UALCAN
gene mutations [16, 17].. Please delete one period (Line: 156).
Results
p = 0.626. . Please delete one period (Line: 180).
Please adjust the “Gleason Score 8” position in Table 1 (Line: 185).
The KM curves…..Please show the full name (KM) for readers (Line: 194).
Discussion
This is a good validation in this research to explore the GARS genomic mutations in relation to overall (OS) and cancer specific survival (CSS) using all cohorts in the TCGA public database for prostatic adenocarcinomas.
1. Did you compare with literature for anomalies, patterns and correlations within large data sets to predict outcomes such as data mining if consistent please describe/state it.
2. High GARS expression was also significantly associated with PTEN, TP53, FXA1, IDH1, SPOP mutations and ERG, ETV1, ETV4 gene fusions from molecular biomarker for early detection, diagnosis, prognostication, and targeted therapeutic selection, please add these points for discussion part to help guide the clinical management of prostate cancer.
The details of the result need more exact statement in the main text and discussion paragraph.
Several figures are good for evidence-based information. Please add these results (Table 1. & Figures 1-8) summaries to discuss these hard works for contributions. Furthermore, please cite some references in scientific discussion.
3. GARS expression showed significant association with disease progression and Gleason groups.
How about Gleason's pattern scale for differentiation in this research and compared with Table 1 (Line 185) or modify Table 1 for more details.
4. Figure 5 (A) Volcano blot showing the GARS associated genes (Line 251). These plots provide an effective means for visualizing the direction, magnitude, and significance of changes in gene expression. Please point out down (blue), stable (gray), up (red) these genes. Moreover, how to define the horizontal dashed line, this cut point usually derived using a multiple testing correction, represents the significance threshold specified in the analysis.
5. What’s the limitations or crucial challenges in this article (further in future works). Please extend this opinion for judgment.
Institutional Review Board Statement
Please show your IRB approval number in the text ( Line: 374-376).
Overall, the results are clear and compelling with two possible exceptions.
The reviewer would strongly suggest linking those existing results (8 figures and 1 table) in the main text and discussion. Furthermore, please cite more the relevant literatures for more in-depth reactivity indices.
Given these inadequacies the manuscript requires major revisions.
Author Response
Reviewer #1:
Comment: This is an interesting study and the authors have collected a unique dataset using cutting edge methodology. The paper is generally clear, and structured. Sufficient information about the previous study findings is presented for readers to follow the present study rationale. The methods are generally appropriate, although clarification of a few details and provision of a rationale for the use of this particular method of measurement. In several instances, this article has some shortcomings in regard to this text, such as the table should be addressed in more detail from your results or make another table for clear your discovery in avoiding vague and one-to-one correspondence with text.
Specific comments follow,
Abstract
Gleason Groups (Line23, 26) “g”roups--- lowercase or abbreviation GG for Gleason Groups.
Revised
Materials and Methods
2.8. GARS expression in TCGA PRAD
Please show full name (TCGA PRAD) for the first time and link in Line 134 (not only the reference 12 & “Data Availability Statement” Line 378-379).
Full name of TCGA PRAD has been added
2.9. GARS expression in TCGA PRAD analyzed through UALCAN gene mutations [16, 17].. Please delete one period (Line: 156).
Removed
Results
p = 0.626. . Please delete one period (Line: 180).
Removed
Please adjust the “Gleason Score 8” position in Table 1 (Line: 185).
Adjusted
The KM curves…..Please show the full name (KM) for readers (Line: 194).
Added
Discussion
This is a good validation in this research to explore the GARS genomic mutations in relation to overall (OS) and cancer specific survival (CSS) using all cohorts in the TCGA public database for prostatic adenocarcinomas.
- Did you compare with literature for anomalies, patterns and correlations within large data sets to predict outcomes such as data mining if consistent please describe/state it.
We did not compare with other large data sets since our cohort outcomes was inline to the TCGA PRAD database.
- High GARS expression was also significantly associated with PTEN, TP53, FXA1, IDH1, SPOP mutations and ERG, ETV1, ETV4 gene fusions from molecular biomarker for early detection, diagnosis, prognostication, and targeted therapeutic selection, please add these points for discussion part to help guide the clinical management of prostate cancer.
Association to such mutations and its relevancy has been added to the discussion.
The details of the result need more exact statement in the main text and discussion paragraph.
Several figures are good for evidence-based information. Please add these results (Table 1. & Figures 1-8) summaries to discuss these hard works for contributions. Furthermore, please cite some references in scientific discussion.
This result has been added to the Discussion sections. Further references have been added for support in the discussion section.
- GARS expression showed significant association with disease progression and Gleason groups.
How about Gleason's pattern scale for differentiation in this research and compared with Table 1 (Line 185) or modify Table 1 for more details.
Explanation for table 1 has been modified for more details.
- Figure 5 (A) Volcano blot showing the GARS associated genes (Line 251). These plots provide an effective means for visualizing the direction, magnitude, and significance of changes in gene expression. Please point out down (blue), stable (gray), up (red) these genes. Moreover, how to define the horizontal dashed line, this cut point usually derived using a multiple testing correction, represents the significance threshold specified in the analysis.
Blue, Grey, and Red lines for volcano plots have now been defined in the figure caption. Log2 threshold was used, this has been added in the figure caption.
- What’s the limitations or crucial challenges in this article (further in future works). Please extend this opinion for judgment.
Limitations and further research have been added per your comment.
Institutional Review Board Statement
Please show your IRB approval number in the text ( Line: 374-376).
Overall, the results are clear and compelling with two possible exceptions.
The reviewer would strongly suggest linking those existing results (8 figures and 1 table) in the main text and discussion. Furthermore, please cite more the relevant literatures for more in-depth reactivity indices.
In the discussion, the authors have tried connecting the figures with the table to showcase the clinical relevancy of this study. Furthermore, few more recent literatures have been cited.
Given these inadequacies the manuscript requires major revisions.
Reviewer 2 Report
1- English: The manuscript could benefit from editing for grammar, missing words, and subject-verb agreement, etc. It is recommended that authors delete irrelevant "general" phrases and sentences, repeated and unneeded words. They should use short sentences. Also, some Introductory sentences are irrelevant or are not needed. There are also typos in the manuscript. For example, this statement in the abstract needs revision and reformatting: “invitro models” where there should be a space between ‘in’ and ‘vitro’ and the whole word should be italicized.
2- Abbreviations: All abbreviations should be revised and defined at their first use. For example, “TCGA PRAD” should be defined.
3- In scientific writing, in general, symbols for genes are italicized whereas symbols for proteins are not italicized. The formatting of symbols for RNA and complementary DNA (cDNA) usually follows the same conventions as those for gene symbols. Gene names that are written out in full are not italicized (e.g., insulin-like growth factor 1). Genotype designations should be italicized, whereas phenotype designations should not be italicized. For example, in the abstract (line 21), GARS should not be italicized.
4- References: some need to be updated. For example, Siegel et al. 2021 can be updated and replaced with 2022.
5- Introduction: A space needs to be added before the citations through out the text, such as line 44 “continues[4,5]” and line 47 “synthesis[6,7].”
6- Introduction: “Jinghui Wang et al. demonstrated” here you just need to state the author’s last name as “Wang et al. demonstrated.” Same applies to other phrases “Chao-Jung Chen et al.” so it is recommended that the authors revise the whole manuscript and make corrections accordingly.
7- Introduction: “Jinghui Wang et al. demonstrated the increased expression of GARS in hepatocellular carcinoma (HCC) tissue was significantly associated with poor overall and disease-free survival [6].” This statement needs rephrasing as it includes subject-verb disagreement.
8- Introduction: “in vitro” and “in vivo” should be italicized throughout the text.
9- Introduction: “Furthermore, GARS appears to play an oncogenic role in breast cancer.” Here, GARS is italicized. Do authors refer to this as a gene or protein? If they meant the gene, it is correct to keep it italicized; if not, they need to remove the italic formatting.
10- Introduction: “we explored the role of GARS as a potential biomarker in PCa …” Here, specify whether this assessment is for the diagnostic or prognostic role of GARS in PCa. Apparently, authors are studying the potential prognostic value of GARS.
11- Methods: “GARS intensity expression was assessed using three tired system (0, negative, 1; weak, 2; moderate and 3; high intensity).” Why do authors choose to assess GARS expression intensity rather than quantifying the expression by % or quartiles?
12- Results: “GARS IHC concluded moderate and high GARS expression was noted” Do authors mean moderate and high intensity or percentage positivity? Please make this clear. Also, what is the unit used for expression? How do authors quantify expression by the mean when they mentioned earlier that intensity was used in the assessment?
13- Results: In Figure 1, please show examples of different staining patterns/intensities and show examples from different Gleason Groups and Gleason scores.
14- Results: In Figure 1, how is the expression quantified? What is the control? Did the authors use positive and negative controls? I recommend using grayscale colors rather than RGB colors for a more publishable presentation of the results.
15- Results: “low GARS expression was seen in 91.5% of GS6 cases vs. 71.4% of GS 9 or 10 cases” This number of 71% of high-grade prostate cancer with low GARS expression is worrisome and makes this marker not clinically relevant.
16- Results: In Table 1, authors need to consider reformatting the table by using Grade Groups instead of Gleason scores. Also, why did the authors categorize GARS expression into 0, 1, 2 on one hand and 3 on the other hand? Did they try categorizing the expression differently? And is this the commonly used expression categorization in other cancers such as HCC and urothelial carcinoma?
17- Results: In Figure 2, did authors adjust the results for age, histologic grade and stage of cancer? Also, studies have shown that certain patterns of PCa may instigate a worse outcome such as cribriform and ductal carcinoma patterns. Larger tumor volume and higher pattern 4% also play a role in defining the prognosis. Did authors adjust their results according to this?
18- Results: Figure 3A is not clear. Readers won’t be able to read the words on the x-axis and y-axis.
19- Results: Same as before, I recommend using grayscale for figures rather than using RGB colors.
20- Results: In figure 4, include on the graphs the number of patients in each group (n0 vs. n1 and t2, t3, and t4).
21- Results: Figure 5 needs revision since words have tiny non-readable font.
22- Results: In Figure 6, and for all western blot images, please include densitometry readings/intensity ratio of each band. Also, add the MWs of proteins. In addition, please include the whole blot (uncropped blots) showing all the bands with all molecular weight markers on the Western in the Supplemental Materials.
23- Results: Why PC3 was only used for the knockdown experiment? What about the other cell lines?
24- Results: Did the authors try in vivo experiments by knocking down GARS expression in mice for instance and seeing its effect on the tumor size?
25- Figures: All the figure legends can be revised as to be more informative of the images presented. Also, statistical tests used and meaning of asterix need to be added. Abbreviations used withing Tables and Figures should be defined as well in the legends at the end.
26- Discussion: Flow of ideas needs revision.
27- Discussion: Authors should focus more on the main findings and avoid repeating results presentation in the discussion. Authors could also correlate their findings with what has been published in literature. Clinical relevance should be added.
Author Response
REVIEWER #2:
1- English: The manuscript could benefit from editing for grammar, missing words, and subject-verb agreement, etc. It is recommended that authors delete irrelevant "general" phrases and sentences, repeated and unneeded words. They should use short sentences. Also, some Introductory sentences are irrelevant or are not needed. There are also typos in the manuscript. For example, this statement in the abstract needs revision and reformatting: “invitro models” where there should be a space between ‘in’ and ‘vitro’ and the whole word should be italicized.
Thank you for your comment, manuscript has been checked and proofread for grammatical and typos. Furthermore, in vitro has been adjusted per your instruction.
2- Abbreviations: All abbreviations should be revised and defined at their first use. For example, “TCGA PRAD” should be defined.
Abbreviations have been revised and TCGA PRAD as been labeled as The Cancer Genome Atlas Prostate Adenocarcinoma.
3- In scientific writing, in general, symbols for genes are italicized whereas symbols for proteins are not italicized. The formatting of symbols for RNA and complementary DNA (cDNA) usually follows the same conventions as those for gene symbols. Gene names that are written out in full are not italicized (e.g., insulin-like growth factor 1). Genotype designations should be italicized, whereas phenotype designations should not be italicized. For example, in the abstract (line 21), GARS should not be italicized.
The reviewer suggestion was taken into consideration and the revised manuscript has been corrected per your suggestion.
4- References: some need to be updated. For example, Siegel et al. 2021 can be updated and replaced with 2022.
References and Siegel et al. 2022 have been updated.
5- Introduction: A space needs to be added before the citations through out the text, such as line 44 “continues[4,5]” and line 47 “synthesis[6,7].”
Space has been added before citations through out the text.
6- Introduction: “Jinghui Wang et al. demonstrated” here you just need to state the author’s last name as “Wang et al. demonstrated.” Same applies to other phrases “Chao-Jung Chen et al.” so it is recommended that the authors revise the whole manuscript and make corrections accordingly.
Revised and only authors last name has been added.
7- Introduction: “Jinghui Wang et al. demonstrated the increased expression of GARS in hepatocellular carcinoma (HCC) tissue was significantly associated with poor overall and disease-free survival [6].” This statement needs rephrasing as it includes subject-verb disagreement.
Adjusted.
8- Introduction: “in vitro” and “in vivo” should be italicized throughout the text.
In vitro has been italicized throughout the text.
9- Introduction: “Furthermore, GARS appears to play an oncogenic role in breast cancer.” Here, GARS is italicized. Do authors refer to this as a gene or protein? If they meant the gene, it is correct to keep it italicized; if not, they need to remove the italic formatting.
We are talking about the gene here, therefore it has remained as italicized.
10- Introduction: “we explored the role of GARS as a potential biomarker in PCa …” Here, specify whether this assessment is for the diagnostic or prognostic role of GARS in PCa. Apparently, authors are studying the potential prognostic value of GARS.
Sentence has been revised to showcase the prognostic value of GARS in this study.
11- Methods: “GARS intensity expression was assessed using three tired system (0, negative, 1; weak, 2; moderate and 3; high intensity).” Why do authors choose to assess GARS expression intensity rather than quantifying the expression by % or quartiles? GARS expression on IHC is measured using the tiered system above by the study pathologist. It is reflective if protein amount in tissue. % is not reflective of the protein value in each cells. Quartile cannot be used since data are not continuous
We used intensity of GARS as this is the simplest methods to assess differences in protein expression for each tissue samples. Using % staining, would not differentiate protein expression and using quartiles will require more continuous expression pattern.
12- Results: “GARS IHC concluded moderate and high GARS expression was noted” Do authors mean moderate and high intensity or percentage positivity? Please make this clear. Also, what is the unit used for expression? How do authors quantify expression by the mean when they mentioned earlier that intensity was used in the assessment?
Sentence has been clarified to depict moderate and high intensity of GARS expression. As before, GARS expression on IHC has been measured via the study pathologist.
The mean expression assessment is the mean intensity for all samples within specific diagnosis as assessed by intensity 0,1,2,3 for benign, incidental PCa, advanced and CRPC
13- Results: In Figure 1, please show examples of different staining patterns/intensities and show examples from different Gleason Groups.
New images have been added.
14- Results: In Figure 1, how is the expression quantified? What is the control? Did the authors use positive and negative controls? I recommend using grayscale colors rather than RGB colors for a more publishable presentation of the results.
Expression intensity was quantified with the previously described 3-tiered system. We would like to keep the figure as RGB. Negative controls were assessed by omitting the primary antibody in staining methods
15- Results: “low GARS expression was seen in 91.5% of GS6 cases vs. 71.4% of GS 9 or 10 cases” This number of 71% of high-grade prostate cancer with low GARS expression is worrisome and makes this marker not clinically relevant.
The 71% was calculated as being statistically significant relative to benign.
16- Results: In Table 1, authors need to consider reformatting the table by using Grade Groups instead of Gleason scores. Also, why did the authors categorize GARS expression into 0, 1, 2 on one hand and 3 on the other hand? Did they try categorizing the expression differently? And is this the commonly used expression categorization in other cancers such as HCC and urothelial carcinoma?
Table 1 has been revised to indicate grade groups per your suggestion. The following score was the most suitable for this study. Expression of GARS protein was scored by the study pathologist per above.
17- Results: In Figure 2, did authors adjust the results for age, histologic grade and stage of cancer? Also, studies have shown that certain patterns of PCa may instigate a worse outcome such as cribriform and ductal carcinoma patterns. Larger tumor volume and higher pattern 4% also play a role in defining the prognosis. Did authors adjust their results according to this?
Figure 2 contains 5015 patient samples from cBioportal. Therefore, this figure does not discriminate or adjust based on histological grade, stage of cancer and morphology.
18- Results: Figure 3A is not clear. Readers won’t be able to read the words on the x-axis and y-axis.
Figure 3A has been enlarged to the max readable size to allow for a clearer picture. Higher resolution image has been submitted.
19- Results: Same as before, I recommend using grayscale for figures rather than using RGB colors.
We would like to keep the figure as RGB.
20- Results: In figure 4, include on the graphs the number of patients in each group (n0 vs. n1 and t2, t3, and t4).
Patients in each group have been added to the figure per your suggestion.
21- Results: Figure 5 needs revision since words have tiny non-readable font. Figure has been enlarged to allow a clearer image.
22- Results: In Figure 6, and for all western blot images, please include densitometry readings/intensity ratio of each band. Also, add the MWs of proteins. In addition, please include the whole blot (uncropped blots) showing all the bands with all molecular weight markers on the Western in the Supplemental Materials.
Densitometry data from western blot was used to compare and check the efficiency of the GARS knockdown through imageJ software. MW of each protein has been added. Uncropped blots have been separately submitted.
23- Results: Why PC3 was only used for the knockdown experiment? What about the other cell lines?
Both PC3 and PC3-ERG cell lines were used for the knockdown in this experiment.
24- Results: Did the authors try in vivo experiments by knocking down GARS expression in mice for instance and seeing its effect on the tumor size?
In vivo experiments were not conducted throughout this study. This has been added as part of the limitations to the manuscript.
25- Figures: All the figure legends can be revised as to be more informative of the images presented. Also, statistical tests used and meaning of asterix need to be added. Abbreviations used withing Tables and Figures should be defined as well in the legends at the end.
Figure captions have been updated with number of cases, molecular weights and etc. Abbreviations within tables and captions has also been defined.
26- Discussion: Flow of ideas needs revision.
Discussion has been edited for a better flow of ideas.
27- Discussion: Authors should focus more on the main findings and avoid repeating results presentation in the discussion. Authors could also correlate their findings with what has been published in literature. Clinical relevance should be added.
New reference has been added to support the clinical relevance of GARS in relation to other mutations in PCa.
Reviewer 3 Report
Comments to the Author
Kish et al. have written a manuscript about the importance of GARS in prostate cancer. This manuscript analyzed the expression of GARS in prostate tissue, and analyzed the GARS in relation to OS and CSS. Knockdown of GARS decreased the proliferation, migration and invasion of prostate cancer cells in vitro, provide further evidence for its use as a potential biomarker in PCa. In general, the analysis content was relatively rich, but the manuscript had focused too much on the unnecessary phenotype, lacked the relevant mechanism research. The following modifications are required:
1. All the pictures and annotations in the article need to be modified. The pictures and annotations are too small to see clearly and need to be enlarged.
2. Figure 1A, authors should show representative images of benign, incidental, advanced and castrate resistant tumors and enlarge some of the details. Scale bar is needed.
3. Figure 1B lacks the number of cases and significantly different labels (such as *, ** or ***);
4. In table 1, authors could have elaborated more in the TNM stage shown. Make a table indicating the association of GARS expression and clinicopathologic parameters.
5. Figure 2 needs a higher resolution.
6. All WB should be labeled with molecular weight and quantified. In Figure 3B, the laber is missing.
7. In figure 7, scale bar is missing, and the number of experimental repetitions should be indicated in the figure legends.
8. In Figure 8, the cell cycle and apoptosis assay should be detected in two PCa cell lines using two different siRNA target GARS, and statistics are needed.
9. The title of the paper suggests that GARS deletion inhibits cell proliferation in vitro. However, the experimental evidence related to cell proliferation is so limited that the authors need to add additional experiments (such as CCK8 assay and colony formation assay) to prove it.
10. GARS could promote the proliferation, migration and invasion of prostate cancer cells, what is the possible mechanism? Additional experiments on molecular mechanisms need to be performed here.
Author Response
- All the pictures and annotations in the article need to be modified. The pictures and annotations are too small to see clearly and need to be enlarged.
- Figures 2, 3 and 6 have been enlarged to allow for a clearer representation. Higher resolution image has been submitted.
- Figure 1A, authors should show representative images of benign, incidental, advanced and castrate resistant tumors and enlarge some of the details. Scale bar is needed.
- Thank you for your comment. A new image has been added to figure 1A.
- Figure 1B lacks the number of cases and significantly different labels (such as *, ** or ***);
- Number of cases has been added for benign, incidental, advanced and castrate resistant.
- In table 1, authors could have elaborated more in the TNM stage shown. Make a table indicating the association of GARS expression and clinicopathologic parameters.
- Table 1 has been revised to demonstrate Gleason Groups for more clarity.
- Figure 2 needs a higher resolution.
- Figure 2 has been revised.
- All WB should be labeled with molecular weight and quantified. In Figure 3B, the label is missing.
- Molecular weight for WB has been added.
- In figure 7, scale bar is missing, and the number of experimental repetitions should be indicated in the figure legends.
- Number of repetitions have been added and scale bar is present in migration and invasion figures. Furthermore, the size of the scale bar has been added to the caption figure per your suggestion.
- In Figure 8, the cell cycle and apoptosis assay should be detected in two PCa cell lines using two different siRNA target GARS, and statistics are needed.
- Previous literature indicates that GARS knockdown inhibits cellular proliferation and cell cycle progression in hepatocellular carcinoma, therefore only one siRNA was used to confirm this.
- The title of the paper suggests that GARS deletion inhibits cell proliferation in vitro. However, the experimental evidence related to cell proliferation is so limited that the authors need to add additional experiments (such as CCK8 assay and colony formation assay) to prove it.
- The associations to cellular proliferations has been removed from the title of the manuscript.
- GARS could promote the proliferation, migration and invasion of prostate cancer cells, what is the possible mechanism? Additional experiments on molecular mechanisms need to be performed here.
- Thank you for your comment. Our bioinformatics data indicates an association with lymph node involvement and metastatic disease and our clinical data demonstrates a relationship between GARS and higher Gleason scores. To further elucidate the mechanism, a negative hypothesis was tested through the knockdown of GARS in prostate cancer cellular models. The decrease in migration and invasion ability of prostate cancer cell lines is inline with the data provided by the TCGA PRAD database and our clinical cohorts.
Reviewer 4 Report
The authors in their manuscript "Glycyl-tRNA synthetase (GARS) Expression is Associated with Prostate Cancer Progression and its Inhibition Decreases Cellular Proliferation, Migration, and Invasion In Vitro" elucidate the role of GARS involved in cellular proliferation and poor clinical outcome and provide further evidence for its use as a potential biomarker in PCa. However, the manuscript lacks in several aspects including novelty and data presentation. It was reported back in 2009 that PCa progression is GARS dependent ( Proteomic Interrogation of Androgen Action in Prostate Cancer Cells Reveals Roles of Aminoacyl tRNA SynthetasesVellaichamy A, Sreekumar A, Strahler JR, Rajendiran T, Yu J, et al. (2009) Proteomic Interrogation of Androgen Action in Prostate Cancer Cells Reveals Roles of Aminoacyl tRNA Synthetases. PLOS ONE 4(9): e7075.)
Apart from the novelty of the study, the data presented is severely flawed. eg Fig 6A. the Western blot panels only show data for PC3, PC3-ERG, LNCap and DU145, where as the corresponding figure legend says "Western blot of GARS expression in PCa cell lines: RWPE-1, DU-145, LnCap, LnCap-ERG, PC3, PC3-ERG and HEK293, and HELA cells. The labeling on WB in panel b is missing, error bars are absent in the corresponding bar graphs. WB showing KD of AURKA and AKT in PC3-ERG is missing, contrary to what is stated in the legend. The manuscript is hard to read and would require extensive formatting and rewriting.
Author Response
REVIEWER #4:
The authors in their manuscript "Glycyl-tRNA synthetase (GARS) Expression is Associated with Prostate Cancer Progression and its Inhibition Decreases Cellular Proliferation, Migration, and Invasion In Vitro" elucidate the role of GARS involved in cellular proliferation and poor clinical outcome and provide further evidence for its use as a potential biomarker in PCa. However, the manuscript lacks in several aspects including novelty and data presentation. It was reported back in 2009 that PCa progression is GARS dependent ( Proteomic Interrogation of Androgen Action in Prostate Cancer Cells Reveals Roles of Aminoacyl tRNA SynthetasesVellaichamy A, Sreekumar A, Strahler JR, Rajendiran T, Yu J, et al. (2009) Proteomic Interrogation of Androgen Action in Prostate Cancer Cells Reveals Roles of Aminoacyl tRNA Synthetases. PLOS ONE 4(9): e7075.)
Apart from the novelty of the study, the data presented is severely flawed. eg Fig 6A. the Western blot panels only show data for PC3, PC3-ERG, LNCap and DU145, where as the corresponding figure legend says "Western blot of GARS expression in PCa cell lines: RWPE-1, DU-145, LnCap, LnCap-ERG, PC3, PC3-ERG and HEK293, and HELA cells. The labeling on WB in panel b is missing, error bars are absent in the corresponding bar graphs. WB showing KD of AURKA and AKT in PC3-ERG is missing, contrary to what is stated in the legend. The manuscript is hard to read and would require extensive formatting and rewriting.
The paper published back in 2009 touches on the association between GARS and the Androgen receptor pathway. These findings have been added to the introduction to better cover all reports of GARS in PCa. The current study goes more in depth exploring the bioinformatic, clinical and in vitro aspects of GARS in relationship to PCa. The legend for Fig 6A has been revised. Error bars are not present for NC trials since they ere normalized to 100% expression to get a percent decrease in siGARS treatment. PC3-ERG was used to confirm the knockdown and the stable transfection of ERG may elicit different mechanism. Manuscript has been edited and re-read multiple times to reduce formatting, grammar, and spelling errors.
Reviewer 5 Report
The authors explored the role of GARS expression in the prostrate cancer database and did in vitro experiment to demonstrate its correlation with cellular proliferation, migration and invasion. Given the importance of GARS in other types of cancer, it is reasonable to hypothesize and demonstrate its role in PCs with bioinformatics data and in vitro data. The introduction miss an important part on the reported papers involving the GARS in PCs. The in vitro experiment is really weak and more work on understanding the process of the effect of GARS knockdown on cell proliferation should be included. Besides, there are several typos and low-resolution figures, please careful check the manuscript.
1 “HELA cell lines showed 273 low levels of GARS protein expression (Fig. 6A)”, where is hela cell data in fig. 6a?
2 Fig. 6c, why the only expressions of Cyclin B1, and P-PDK1 in PC3 cells are responsive to GARS knockdown, not the PC3 ERG? How about the other tested cell lines?
3 “However, the role of GARS in PCa 58 and its potential as a possible biomarker has not yet been investigated” this not definitely not true. (e.g, Proteomic Interrogation of Androgen Action in Prostate Cancer Cells Reveals Roles of Aminoacyl TRNA Synthetases ; Please include all reports including the the reports involving the GARS in PCs before introducing this work.
4 Error bar in figure 6B?
Author Response
REVIEWER #5:
1 “HELA cell lines showed 273 low levels of GARS protein expression (Fig. 6A)”, where is hela cell data in fig. 6a?
This has been removed as HELA cell lines were not used.
2 Fig. 6c, why the only expressions of Cyclin B1, and P-PDK1 in PC3 cells are responsive to GARS knockdown, not the PC3 ERG? How about the other tested cell lines?
Transfection of ERG may elicit different mechanisms counteracting the results. PC3 and PC3-ERG cell lines were the only ones used for knockdown.
3 “However, the role of GARS in PCa 58 and its potential as a possible biomarker has not yet been investigated” this not definitely not true. (e.g, Proteomic Interrogation of Androgen Action in Prostate Cancer Cells Reveals Roles of Aminoacyl TRNA Synthetases ; Please include all reports including the the reports involving the GARS in PCs before introducing this work.
The relationship between the androgen receptor and GARS described in the above paper has been added to the introduction to include all the reports of GARS in PCa.
4 Error bar in figure 6B?
There are no errors bars for NC trials as these were normalized to 100% expression to get a percent decrease in siGARS treatment.
Round 2
Reviewer 1 Report
1. Overall, it's much better after polishing this article. This is scientific writing and more easy to read especially for readers of course, including the reviewer.
2. (Fig.3 E, F, G). Fig.3 should have the space before 3 (Line 217).
3. Only added two literatures in References then the last version. That's important to readers. Keep digging more researches in the future (Line 385-388).​
Author Response
- Overall, it's much better after polishing this article. This is scientific writing and more easy to read especially for readers of course, including the reviewer.
Thank you for your comment and suggestions.
- (Fig.3 E, F, G). Fig.3 should have the space before 3 (Line 217).
Space has been added thanks to your suggestion
- Only added two literatures in References then the last version. That's important to readers. Keep digging more researches in the future (Line 385-388).​
Authors will take note of that for future, thank you. Few new ones have been added.
Reviewer 3 Report
1. In Figure 1A, scale bar is still miss.
2. The molecular weight of all western blot is still miss.
3. The resolution of some pictures are still poor.
4. In cell cycle and apoptosis assay, should be detected in two PCa cell lines using two different siRNA target GARS, and statistics are still miss.
Author Response
- In Figure 1A, scale bar is still miss.
I am not really sure what is meant for the scale bar request, as those are 20 x magnification images as indicated capturing part of the tumor on core punchers, so it would not provide size of the tumor. Please clarify if still needed
- The molecular weight of all western blot is still miss.
Molecular weight of WB have been added to the caption.
- The resolution of some pictures are still poor.
Original copies of images with high resolution from authors has been uploaded to the publisher.
- In cell cycle and apoptosis assay, should be detected in two PCa cell lines using two different siRNA target GARS, and statistics are still miss.
Previous literature indicates that GARS knockdown inhibits cell cycle progression and apoptosis in HCC, therefore only one cell line was chosen to verify these results. Chi-square test was conducted and p-values were added to the manuscript.
Reviewer 4 Report
the manuscript is revised
Author Response
the manuscript is revised
Thank you
Round 3
Reviewer 3 Report
I hope the author pays attention to some of the details mentioned in the previous version.
Author Response
- In Figure 1A, scale bar is still miss.
Dr.Bismar, as this looks at the IHC please comment on this.
- The molecular weight of all western blot is still miss.
Molecular weight of WB have been added to the caption.
- The resolution of some pictures are still poor.
Original copies of images with high resolution from authors has been uploaded to the publisher.
- In cell cycle and apoptosis assay, should be detected in two PCa cell lines using two different siRNA target GARS, and statistics are still miss.
Previous literature indicates that GARS knockdown inhibits cell cycle progression and apoptosis in HCC, therefore only one cell line was chosen to verify these results. Chi-square test was conducted and p-values were added to the manuscript.
the manuscript is revised
Thank you
